# Globoside Is an Essential Intracellular Factor Required for Parvovirus B19 Endosomal Escape

**DOI:** 10.3390/cells13151254

**Published:** 2024-07-25

**Authors:** Jan Bieri, Corinne Suter, Oliver Caliaro, Seraina Bartetzko, Cornelia Bircher, Carlos Ros

**Affiliations:** 1Department of Chemistry, Biochemistry and Pharmaceutical Sciences, University of Bern, Freiestrasse 3, 3012 Bern, Switzerland; 2Graduate School for Cellular and Biomedical Sciences, University of Bern, Mittelstrasse 43, 3012 Bern, Switzerland

**Keywords:** parvovirus B19, B19V, globoside, virus entry, endosomal escape, phospholipase A2

## Abstract

Human parvovirus B19 (B19V), like most parvoviruses, possesses phospholipase A2 (PLA2) activity, which is thought to mediate endosomal escape by membrane disruption. Here, we challenge this model and find evidence for a mechanism of B19V entry mediated by the glycosphingolipid globoside without endosome disruption and retrograde transport to the Golgi. We show that B19V PLA2 activity requires specific calcium levels and pH conditions that are not optimal in endosomes. Accordingly, endosomal membrane integrity was maintained during B19V entry. Furthermore, endosomes remained intact when loaded with MS2 bacteriophage particles pseudotyped with multiple B19V PLA2 subunits, providing superior enzymatic potential compared to native B19V. In globoside knockout cells, incoming viruses are arrested in the endosomal compartment and the infection is blocked. Infection can be rescued by promoting endosomal leakage with polyethyleneimine (PEI), demonstrating the essential role of globoside in facilitating endosomal escape. Incoming virus colocalizes with Golgi markers and interfering with Golgi function blocks infection, suggesting that globoside-mediated entry involves the Golgi compartment, which provides conditions favorable for the lipolytic PLA2. Our study challenges the current model of B19V entry and identifies globoside as an essential intracellular receptor required for endosomal escape.

## 1. Introduction

The *Parvoviridae* family comprises an exceptionally diverse group of viruses that infect a wide range of hosts. Relevant members of this family include human parvovirus B19 (B19V) and adeno-associated virus (AAV), a leading platform for gene therapy [1]. B19V is a highly prevalent human pathogen that is primarily transmitted via the respiratory route [2,3]. The small, non-enveloped icosahedral capsid of B19V is composed of VP1 and VP2 proteins. VP1 is identical to VP2 except for 227 additional amino acids at the N-terminus, the so-called VP1 unique region or VP1u [4,5]. Infection in healthy children can cause a skin rash known as *erythema infectiosum* [6]. Polyarthropathy is often associated with the infection in healthy adults, transient aplastic crisis in individuals with hemolytic disorders, and chronic infection and anemia in immunocompromised hosts [7]. In non-immune pregnant women, the virus can infect the developing fetus, potentially leading to *hydrops fetalis* [8].

Understanding the mechanisms by which parvoviruses enter the cell is crucial for developing strategies to prevent and treat the associated diseases, but also to improve gene delivery by AAV vectors. Parvoviruses have evolved to infect a wide variety of hosts and tissues. Despite their diversity, the mechanism of entry into host cells shares common features. Parvovirus uptake is mediated by different receptors and co-receptors, which determine their cell and tissue tropism. Upon receptor interaction, parvoviruses are primarily internalized via clathrin-mediated endocytosis [9,10,11] and transported to early endosomes, where they are exposed to a low pH environment. The acidic conditions within the endosomes induce changes in the structure of the viral capsid, leading to the exposure of the VP1u region and its constitutive phospholipase A2 (PLA2) domain [12,13,14,15]. The viral PLA2 activity can break down phospholipids present in membranes. Although there is no direct evidence, the current model of parvovirus entry suggests that the capsid-tethered PLA2 enzyme allows the endosomal membrane to be breached, enabling the release of incoming viral particles into the cytoplasm [14,16]. However, growing evidence from studies involving AAV suggests that following uptake by endocytosis, incoming viruses undergo retrograde transport to the Golgi and subsequent nuclear entry through mechanisms that remain poorly understood [17,18,19,20,21]. The pan-serotype receptor AAVR and the Golgi-localized G protein-coupled receptor GPR108 are essential for the translocation of incoming AAV from endosomes to the Golgi, although the specific mechanism is not known [20,22]. Accordingly, the Golgi apparatus appears to be a critical compartment in the intracellular route of AAV from the plasma membrane to the nucleus.

In previous studies, we have shown that B19V infection is controlled by two distinct receptors. Entry through the airway epithelium is mediated by the interaction of B19V with the glycosphingolipid globoside, which is expressed in the ciliated population of the epithelium. This interaction occurs exclusively under the natural acidic conditions of the upper respiratory tract, resulting in virus transcytosis and release at the basolateral side where the pH is neutral [23]. After penetrating the airway epithelium, B19V targets and productively replicates in erythroid progenitor cells (EPCs) in the bone marrow. Although EPCs express globoside, the extracellular neutral pH prevents viral interaction with the glycosphingolipid. Virus uptake into EPCs is mediated by another receptor, here named VP1uR, because the receptor binding domain (RBD) is located at the N-terminus of VP1u [24,25,26]. In addition to its function as a receptor for respiratory entry of B19V, globoside has another essential function after viral entry into EPCs. In globoside knockout (KO) erythroid cells, viral particles are internalized through interaction with VP1uR [27], but they become arrested in the endosomes and the infection is blocked, suggesting a possible role for globoside in endosomal escape [28]. 

Building on our previous research, we investigated the function of globoside in the intracellular trafficking of B19V. Our findings challenge the current model of parvovirus endosomal escape based on membrane disruption by the lipolytic activity of the viral PLA2. We present evidence that low endosomal pH promotes receptor switching from the surface receptor VP1uR, which mediates virus uptake, to the glycosphingolipid globoside. The interaction of incoming viruses with globoside in the acidic endosomes is essential for endosomal escape. In the absence of globoside, infection was rescued by forcing endosomal escape through polyethyleneimine (PEI)-mediated endosome leakage. The endosomal environment is not favorable for the enzymatic activity of the parvoviral PLA2. Accordingly, globoside-mediated entry does not compromise endosomal integrity and involves the Golgi compartment, which provides a better environment for the viral PLA2. This study underscores the pivotal role of globoside as an essential intracellular factor required for infectious entry and provides further insight into the mechanisms behind the restricted tropism of B19V.

## 2. Materials and Methods

### 2.1. Cells and Viruses 

UT7/Epo cells were kindly provided by E. Morita (Tohoku University School of Medicine, Japan) and cultured in DMEM:F12 containing 5% FCS, 100 U/mL of penicillin/streptomycin, and 2 U/mL erythropoietin (Epo). A9 cells were obtained from the American Type Culture Collection (ATCC), cultured in DMEM high glucose with 5% FCS, 50 U/mL of penicillin/streptomycin. Red blood cells (RBCs) were obtained from anonymous blood donations in EDTA-treated tubes (Swiss Transfusion SRC, Bern, Switzerland). B19V was obtained from seronegative infected plasma samples (CSL Behring AG, Charlotte, NC, USA) and depleted of IgGs using sepharose beads. MVM was obtained from ATCC and propagated in A9 cells. PP7 bacteriophages were obtained from ATCC, propagated in *Pseudomonas aeruginosa*, and modified as previously indicated [29].

### 2.2. Recombinant Protein Expression and Click Chemistry 

MS2 bacteriophage virus-like particles (VLPs) and parvovirus VP1u proteins were expressed, purified, and linked as described previously [28]. In brief, assembled capsids were purified from bacteria by freeze-and-thaw followed by ultracentrifugation through a 20% sucrose solution in PBS at 150,000 × *g* for 2 h. The pellet containing the VLPs was resuspended in PBS and capsids were fluorescently labeled with a 50-fold molar excess of NHS–ester modified atto-488 (Atto-Tec, Siegen, Germany). Recombinant VP1u proteins were purified using Ni-NTA beads and B19-VP1u molecules were crosslinked to fluorescently labelled MS2-VLPs or PP7 using a maleimide–NHS–ester bifunction linker (22114, Thermo Fisher Scientific, Waltham, MA, USA).

### 2.3. Drugs and Reagents 

All drugs were purchased from Sigma-Aldrich (St. Louis, MI, USA). Golgicide A (G0923) and Retro-2 (SML1085) were resuspended in DMSO to a final concentration of 20 mM. Brefeldin A (B6542) was dissolved in DMSO at 5 mg/mL. Linear polyethyleneimine (PEI) (PolySciences, Warrington, PA, USA) with a molecular weight of 25 kDa was dissolved in dH_2_O to 1 mM. Bafilomycin (SML1661, Sigma-Aldrich) was obtained as a ready-made solution. The 3 kDa tetramethylrhodamine-labeled dextran was purchased from Invitrogen (D3307, Carlsbad, CA, USA) and was dissolved in dH_2_O to a final concentration of 20 mg/mL. 

### 2.4. Antibodies 

The rabbit monoclonal antibodies (mAb) GM130 (ab52649) against the cis-Golgi and TGN46 (ab50595) against the trans-Golgi network, the mouse mAb Giantin (ab37266) against the Golgi complex, and the rabbit polyclonal antibody (pAb) against GAPDH (ab9485) were purchased from Abcam (Cambridge, UK). The human mAb 860-55D against intact capsids was purchased from Mikrogen (Neuried, Germany) and detected using the pAb Alexa Fluor^®^ 488 conjugated goat anti-human IgG (A11013, Thermo Fisher Scientific). The mouse mAb 3113-81C against viral capsid proteins was obtained from US Biological (Boston, MA, USA). A rabbit pAb against MS2 capsid proteins was purchased from Merck Millipore (ABE76-I, Darmstadt, Germany). The mouse mAb B7 against MVM was kindly provided by J.M. Almendral (Universidad Autónoma de Madrid, Spain). The mouse mAb against Rab9 (MA3-067) was obtained from Thermo Fisher Scientific. Golgi-markers were detected using the pAb Alexa Fluor^®^ 594 conjugated goat anti-rabbit IgG (ab150080, Abcam) and goat anti-mouse IgG (A11005, Thermo Fisher Scientific). MVM was visualized with a pAb Alexa Fluor^®^ 488 conjugated goat anti-mouse (A11001, Thermo Fisher Scientific). MS2 capsid proteins and Rab9 were detected with goat polyclonal anti-mouse immunoglobulin/HRP (P0447), while GAPDH was detected with goat polyclonal anti-rabbit immunoglobulin/HRP (P0448). The HRP-conjugated antibodies were purchased from Agilent Technologies (Santa Clara, CA, USA).

### 2.5. Binding and Internalization Assays 

Binding of recombinant VP1u (100 ng), B19V (10^5^ geq/cell), and PP7-VP1u (10^5^ geq/cell) was carried out at 4 °C for 1 h in UT7/Epo cells resuspended in PBS (pH 7.2) or citrate buffer of desired pH (pH 5.0–6.6, 20 mM citrate buffer, 123 mM NaCl, 2.5 mM KCl). Cells were washed three times with ice-cold buffer and optionally treated with trypsin at 37 °C for 4 min before fixation for immunofluorescence or extraction of DNA or RNA. Internalization of recombinant VP1u (100 ng), B19V (10^5^ geq/cell), MS2-VP1u (10^6^ geq/cell) and MVM (2 × 10^3^ geq/cell) was carried out at 37 °C in UT7/Epo or A9 cells. After 30 min, cells were washed three times with PBS and prepared for immunofluorescence or DNA extraction. To test B19V binding affinity to globoside, red blood cells (RBCs) were washed and resuspended in PiBS (20 mM PIPES pH 7.2–5.6, 121 mM NaCl, 2.5 mM KCl) to yield a 0.5% (*v*/*v*) suspension followed by addition of 3 × 10^9^ purified B19V particles. Binding of viruses to cells was allowed at room temperature for 1 h. Unbound capsids were removed by three subsequent washing steps with buffers of the same pH as used for the binding, followed by DNA extraction. DNA extraction was performed using the GenCatch Plus Genomic DNA Miniprep Kit (1660250, Epoch Life Science, Missouri City, TX, USA) according to the manufacturer’s instructions, and the DNA was amplified in a qPCR assay using the following primers: B19-F: 5′-GGGGCAGCATGTGTTAA-3′; B19-R: 5′-AGTGTTCCAGTATATGGCATGG-3′; MVM-F: 5′-GACGCACAGAAAGAGAGTAACCAA-3′; MVM-R: 5′-CCAACCATCTGCTCCAGTAAACAT-3′. Viral RNA was isolated from PP7 capsids using the QIAmp viral RNA mini Kit (52904, Qiagen, Hilden, Germany) and quantified as previously described [29]. 

### 2.6. Immunofluorescence 

UT7/Epo and A9 cells were fixed with methanol:acetone (1:1) at −20 °C for 4 min. Cells were blocked post-fixation with 10% goat-serum in PBS followed by staining with primary antibodies. Samples were subsequently washed with PBS and bound primary antibodies were detected with Alexa Fluor^®^ dyes conjugated to secondary antibodies. Images were obtained using confocal microscopy (LSM880, Zeiss, Oberkochen, Germany). 

### 2.7. Infectivity Assays 

UT7/Epo and A9 cells were infected with B19V (10^3^–10^5^ geq/cell) and MVM (2 × 10^4^ geq/cell), respectively, at 37 °C for 30 min. Cells were washed followed by extraction of total RNA or incubated further at 37 °C. If drugs were used, cells were pretreated at 37 °C for 15 min prior to addition of the virus. When investigating the influence of PEI on the infection, desired concentrations ranging from 0.1–20 μM were added to cells for 30 min at the same time as the virus. At desired times post-infection, cells were harvested and washed twice with PBS, followed by extraction of total RNA using the GenCatch Total RNA Mini Prep Kit (1660250, Epoch Life Science, Missouri City, TX, USA). RT-qPCR was performed using the primers described above. 

### 2.8. PLA2 Assay

Phospholipase activity was assayed using the sPLA2 Assay Kit (765001, Cayman chemicals, Ann Arbor, MI, USA), according to the manufacturers protocol with the following changes. The assay buffer was prepared without CaCl_2_. A solution of CaCl_2_ (1 M) was used to reach desired calcium concentrations for each individual reaction. The final pH of the reaction was adjusted to fit the desired criteria. For each reaction, either recombinant VP1u (100–1000 ng), MS2-VP1u (5 × 10^10^), B19V (5 × 10^10^), or MVM (10^10^) was assayed. B19V and MVM were heated to 85 °C for 5 min prior to analysis to expose the PLA2 motif. The PLA2 activity was analyzed through the change in absorption at 414 nm each minute for 15 min using a microplate reader. 

### 2.9. Rab9 Knockdown 

Stealth siRNAs against Rab9 mRNA or scrambled siRNA controls were purchased from Invitrogen. UT7/Epo or A9 cells were transfected using Lipofectamine RNAiMAX according to manufacturer’s protocol. In brief, 0.5–2 × 10^5^ cells were washed and resuspended in Opti-MEM. A volume of 1 µL Lipofectamine RNAiMAX were mixed with 10 nM siRNA in Opti-MEM, incubated for 10 min at RT, and added dropwise to the cells. After 2 h at 37 °C, the medium was carefully replaced with fresh culture medium, and cells were incubated at 37 °C for two days. Knockdown efficiency was determined by Western blotting with specific Rab9 and GAPDH antibodies and by RT-qPCR with Rab9 mRNA-specific primers: Rab-F: 5′-GGACAACGGCGACTATCCTT-3′, Rab-9-R: 5′-TGAGCTAGGCTTGGGCTT TC-3′. Infection of transfected cells was carried out as described above. 

### 2.10. Western Blot

MS2-VLPs or lysates of cells used for Rab9-knockdown were separated on a 10% SDS-PAGE gel followed by transfer to a PVDF membrane. After blocking with 5% milk in TBS-T (0.05% Tween^®^ 20) the membrane was incubated with a primary antibody concentrated 1:2000 followed by washing with TBS-T. Chemiluminescence detection using an HRP-conjugated antibody was used to visualize the proteins. 

### 2.11. Statistical Analysis 

GraphPad Prism version 8.0 was used for analysis and visualization of data. Student’s *t*-test was performed to evaluate the statistical significance and was indicated as follows: ns, not significant; * *p* < 0.05; ** *p* < 0.01; *** *p* < 0.001; **** *p* < 0.0001.

## 3. Results

### 3.1. Interaction of B19V with VP1uR and Globoside Receptors Is Modulated by the pH

In a previous study, we found that B19V interacts with the glycosphingolipid globoside exclusively at mildly acidic pH (≈ 6.3) and that this interaction is required for the infection [28]. Globoside and VP1uR are both expressed in erythroid progenitor cells (EPCs) [25,27]. However, B19V does not bind to globoside exposed on the plasma membrane of EPCs because the interaction requires acidic conditions [28]. Accordingly, globoside does not play a role in virus binding and uptake, which is mediated by VP1uR. To investigate the effect of pH on the interaction of B19V with VP1uR, cells were incubated with recombinant VP1u under neutral (pH 7.2) or acidic (pH 5.0) conditions at 4 °C to allow binding and at 37 °C to allow internalization. Treatment of the cells with trypsin removed bound but not internalized proteins. As shown in Figure 1A, binding of recombinant VP1u was significantly reduced and internalization was abolished at low pH. VP1u bound at pH 7.2 was removed by exposing the cells to pH 5.0, indicating that VP1u dissociates from VP1uR when exposed to acidic conditions (Figure 1B).

Bioconjugation of VP1u to the surface of *Pseudomonas aeruginosa* bacteriophage PP7 by click chemistry was performed as previously described [29]. The bioconjugated VP1u behaves identically to the native peptide and confers the ability to interact with VP1uR. This approach allowed for a more sensitive and quantitative determination of the pH-mediated interaction and dissociation of VP1u with VP1uR. UT7/Epo cells were incubated with PP7-VP1u bioconjugates at various pH conditions at 4 °C. Subsequently, cells were washed, and the amount of phage RNA was quantified by RT-qPCR as previously described [29]. To quantify the interaction of B19V with globoside, purified B19V from infected plasma was incubated with red blood cells (RBCs), which express globoside but not VP1uR. RBCs were incubated with viruses at various pH conditions at 4 °C, washed, and the amount of bound B19V quantified by qPCR. VP1u binding to VP1uR is optimal at neutral pH (7.2) and gradually decreases as the pH becomes more acidic. In contrast, the affinity of B19V for globoside increases drastically as the pH decreases (Figure 1C). These results demonstrate the critical role of pH in regulating the affinity of B19V for the two receptors. In the extracellular environment exposed to neutral pH conditions, B19V targets the VP1uR for viral uptake. However, after uptake, when the virus encounters the acidic pH in endosomes, the affinity of incoming viruses for VP1uR decreases, while it increases for globoside. Accordingly, the endosomal compartment provides the conditions for globoside interaction.

### 3.2. Promoting Endosomal Leakage with Polyethyleneimine (PEI) Rescues B19V Infection in Globoside Knockout Cells

Previous studies have shown that in globoside KO UT7/Epo cells, B19V is internalized normally, but incoming particles are trapped in endosomes and infection is blocked [20,21]. We hypothesized that if the essential function of globoside is to facilitate endosomal escape, then promoting endosome-to-cytosol delivery using polyethyleneimine (PEI) could rescue the infection in globoside KO cells. PEI acts as a proton sponge in acidic endosomes, causing osmotic swelling and endosome disruption [30,31].

Globoside KO UT7/Epo cells were infected to allow accumulation of incoming virus within endosomes. B19V was able to internalize globoside KO cells similarly to wild-type (WT) cells, with no detectable differences in intracellular distribution (Figure 2A) and quantity (Figure 2B). However, regardless of the MOI, the infection was blocked in globoside KO cells (Figure 2C). These results confirmed our previous observations that in the absence of globoside, B19V uptake and endosomal accumulation occur as in WT cells, but infection is blocked [28]. To verify the effect of PEI on endosomal integrity, cells were incubated with increasing doses of PEI in the presence of low molecular weight tetramethylrhodamine-labeled dextran (3 kDa) at 37 °C for 30 min. Doses above 1 μM PEI caused significant disruption of endosomes with the release and accumulation of dextran in the cytoplasm and nuclei (Figure 2D). Infection of WT UT7/Epo cells in the presence of PEI had an enhancing effect on the infection at a dose of 1 μM. As expected, the beneficial effect of PEI decreased at doses above 5 μM, which are known to initiate mitochondrially mediated apoptosis [32] (Figure 2E). In globoside KO cells, PEI allowed the virus to circumvent the globoside block and to initiate the infection (Figure 2F). Collectively, these results demonstrate that the primary role of globoside in B19V infection of erythroid progenitor cells is to facilitate endosomal escape of the incoming virus, with no discernible involvement in other steps that precede or follow this crucial process.

### 3.3. The Endocytic Compartment Provides Optimal Conditions for Receptor Switching But Not for the Parvoviral PLA2 Activity

During the entry process, parvoviruses must escape from the endosomes. Like most parvoviruses, the VP1u of B19V harbors a PLA2 enzymatic activity, which has been shown to be crucial for the infection [13]. The lipolytic activity of the viral PLA2 is thought to disrupt the endosomal membrane and facilitate endosomal escape [14]. However, evidence for parvoviral PLA2-mediated endosomal disruption has never been demonstrated. 

The enzymatic activity of the parvoviral PLA2 requires specific conditions [33]. Here, we investigated the calcium and pH requirements of B19V PLA2. As shown in Figure 3A, the enzyme activity is strongly influenced by pH and calcium concentration. The activity is optimal at pH 6.2 and becomes nearly undetectable at acidic (5.0–5.5) and at neutral (7.2) pH. Calcium concentration is also critical, and below 100 μM, the activity is barely detectable. 

The pH-dependent activity of B19V PLA2 was compared with that of pig-tailed macaque parvovirus (PmPV), an erythroparvovirus closely related to B19V [34], and minute virus of mice (MVM), a phylogenetically distant and well-characterized parvovirus model. To this end, the VP1u of PmPV was expressed and purified as previously described [29]. The PLA2 domain of purified MVM capsids was rendered accessible by incubating the capsids at 85 °C for 5 min [35]. Similar to B19V, the PLA2 activity of PmPV and MVM was optimal at mildly acidic pH ≈ 6.2 and strongly inhibited at pH < 5.5. In contrast to B19V and PmPV, the PLA2 from MVM was active at neutral pH (Figure 3B). 

The pH and calcium concentration in the endocytic compartment differ substantially from those in the extracellular milieu. After internalization from the calcium-rich (mM range) extracellular milieu, the concentration in a newly formed endosomal vesicles rapidly decreases to <5 µM, concomitant with the acidification of the organelle [36]. Calcium concentration increases again as endosomes progress through the endo-lysosomal pathway [37]. The mildly acidic pH ≈ 6.2 in early endosomes decreases to pH < 5.5 in late endosomes/lysosomes. Accordingly, the endocytic compartment does not meet the specific pH and calcium concentration required for the lipolytic activity of the parvoviral PLA2. The Golgi apparatus, with an average luminal pH of 6.0–6.7 and a calcium concentration of 130–250 µM [38,39], is the organelle that most closely matches the conditions required for the parvoviral PLA2 activity (Figure 3C).

### 3.4. The Escape of Incoming B19V from Endosomes Does Not Compromise the Integrity of the Endosomal Membranes

Endosomes, with their characteristic pH and calcium concentration, do not provide optimal enzymatic conditions for efficient lipolytic activity of the viral PLA2 enzyme. Therefore, the possibility that the virus uses its PLA2 activity within the endosome to disrupt the endosomal membrane and escape into the cytosol seems unlikely. To confirm this, the integrity of endosomal membranes during B19V entry was examined. UT7/Epo cells were incubated with B19V and low molecular weight tetramethylrhodamine-labeled dextran (3 kDa). Dextrans are rapidly internalized and accumulate in endosomes. However, when the integrity of the endosomal membranes is compromised, the small dextrans escape from the endosomes, resulting in staining of both the cytosol and the nucleus. Mock-infected cells were used as a control. After 30 min, the cells were washed, trypsinized to remove bound but not internalized virus, and processed for immunofluorescence to confirm internalization of B19V. At progressive times after internalization, dextran release from endosomes into the cytosol was examined by confocal microscopy. Dextran release was not detected at any time point, and their endocytic accumulation resembled that of mock-infected cells (Figure 4A). This result suggests that endosomal escape of incoming B19V occurs without causing membrane damage. 

Native B19V typically harbors an average of three VP1u residues per capsid, which is less than the 5–10 copies per virion observed in other parvoviruses [1,40]. We sought to increase the PLA2 activity of B19V and test its effect on the endocytic vesicles. To this end, B19V VP1u subunits were incorporated into bacteriophage MS2 capsids, which do not interact with eukaryotic cells but have similar features to B19V, such as an icosahedral capsid of similar size [41,42]. Recombinant MS2 particles were labeled with the fluorescent dye Atto488 and bioconjugated with multiple full-length B19V VP1u containing the RBD and PLA2 domain (MS2-VP1u) or with a truncated VP1u construct without the PLA2 (MS2-∆C128) (Figure 4B). Western Blot analysis showed that the two VP1u constructs were successfully linked to MS2 capsids (≈ 15–18 per capsid) (Figure 4C). The resulting capsid constructs are schematically outlined in Figure 4D. The PLA2 activity of recombinant MS2-VP1u, native B19V, and MS2-∆C128 was quantitatively assessed. The PLA2 activity of MS2-VP1u was approximately 11 times higher than that of native B19V capsids. As expected, the activity of MS2-∆C128 was undetectable (Figure 4E). Both MS2 constructs were able to internalize UT7/Epo cells and accumulate in the endocytic pathway without noticeable differences (Figure 4F). UT7/Epo cells were incubated with MS2-VP1u or with MS2-∆C128 and tetramethylrhodamine-labeled dextran (3 kDa) at 37 °C for 30 min, washed to remove unbound particles, and further incubated for 1h. Despite the increased PLA2 potential, the accumulation of MS2-VP1u particles in the endosomes did not cause dextran release, indicating that the endosomal membrane remains intact (Figure 4F).

### 3.5. B19V Entry Involves a Functional Golgi Apparatus

The integrity of the endosomal membrane was not compromised by the PLA2 activity of the incoming virus, and increasing the PLA2 potential did not alter this result. The fact that conditions inside endosomes are not favorable for the enzymatic activity of viral PLA2 may explain this observation. These conditions are more favorable in the Golgi apparatus, where the lipolytic activity of viral PLA2 could facilitate virus release into the cytosol.

To investigate the involvement of the Golgi apparatus in B19V infection, brefeldin A (BFA) and golgicide A (GCA) were used, which are known to partially or completely disassemble the Golgi apparatus [43,44]. The effect of BFA and GCA on the morphology of the Golgi apparatus in UT7/Epo cells was confirmed by immunofluorescence staining with giantin, which targets the inter-cisternal cross-bridges of the Golgi complex [45]. Both drugs dispersed the Golgi compartment in UT7/Epo cells within 30 min (Figure 5A). Cells pretreated with either BFA or GCA for 30 min prior to infection with B19V showed a dose-dependent reduction in infectivity as measured by NS1 mRNA quantification. A dose-dependent reduction in infectivity was observed for both drugs (Figure 5B,C). Although the mechanism of Golgi disassembly by BFA and GCA differs, both drugs resulted in a similar reduction in B19V infectivity when applied early in the infection. BFA and GCA showed a similar loss of efficacy against infection when applied at increasing times after virus internalization. The compounds had a significant effect on the infection when applied during the first two hours post-infection (hpi), and both had a minor effect when applied 4 hpi (Figure 5D). This result confirms that the effect on the infectivity is not due to cytotoxicity but rather to the disruption of the infectious entry pathway of the virus. 

The reduction in B19V infectivity after treatment with either BFA or GCA suggests that the functionality of the Golgi compartment is required for entry. In an attempt to circumvent the Golgi compartment during B19V entry, cells treated with GCA were infected in the presence of PEI to permeabilize the endosomes, enabling B19V to exit directly from the endosomal compartment into the cytosol. As shown in Figure 5E, PEI treatment rescued infection in BFA-treated cells, indicating that B19V requires the Golgi compartment for entry and not for subsequent steps of the infection. Similarly, the infection was rescued when PEI was applied to cells treated with GCA (Figure 5F).

The results with Golgi disrupting drugs do not definitively establish whether the virus is trafficked to the Golgi compartment or whether the Golgi apparatus indirectly influences the entry pathway. We next tested the presence of incoming B19V in the Golgi compartment using specific antibodies. Immunofluorescence staining was performed with a conformational anti-B19V capsid antibody (860-55D) and the Golgi markers TGN46, which is mainly localized in the trans-Golgi network (TGN), giantin, a membrane-inserted component of the cis and medial Golgi, and GM130, for the cis-Golgi. The signal of incoming virus did not colocalize with giantin, showed weak association with TGN46, and displayed significant overlap with the cis-Golgi marker GM130. The colocalization signal concerned the body of the cis-Golgi, as well as small vesicles (Figure 5G). This result confirms that the Golgi apparatus is targeted by incoming viruses during the entry process.

### 3.6. B19V Does Not Employ Conventional Retrograde Endosome-to-Golgi Transport

The requirement for a functional Golgi early after virus internalization and the co-localization of viruses with Golgi markers, suggest that the incoming virus is translocated from endosomes to the Golgi as part of the intracellular traffic to the nucleus. Retrograde transport from endosomes to the Golgi involves the retromer complex, which is responsible for the transport of proteins from early endosomes to the Golgi [46]. Retro-2 is a chemical inhibitor that interferes with the function of the retromer complex and inhibits early endosome-to-Golgi transport [47]. UT7/Epo cells were pretreated with increasing doses of Retro-2 and infected with B19V. After 24 h, NS1 mRNA was quantified by RT-qPCR. As shown in Figure 6A, Retro-2 had no effect on B19V infection, indicating that the virus does not use this pathway to enter cells. 

Endosome-to-Golgi retrograde transport occurs not only from early endosomes but also from late endosomes [46]. Rab9 has been shown to localize to tubular late endosomes and is required for the formation and motility of transport vesicles from late endosomes to the Golgi [48]. To investigate the involvement of this pathway in B19V entry, Rab9 was knocked down by RNAi in UT7/Epo cells. The reduction of Rab9 expression was confirmed by Western blot (Figure 6B) and Rab9 mRNA quantification (Figure 6C). The Rab9-knockdown UT7/Epo cells were infected and NS1 mRNA was quantified 24 hpi by RT-qPCR. As shown in Figure 6D, Rab9 knockdown had no effect on infection. Collectively, these results suggest that B19V entry does not involve the canonical endosome-to-Golgi transport.

### 3.7. B19V and the Model Parvovirus MVM Share Common Features in Cell Entry

In this study, we provide evidence that the endocytic environment is incompatible with the enzymatic requirements of the PLA2 of B19V, but also of PmPV and MVM (Figure 3). Therefore, it is plausible that MVM, similar to B19V, does not compromise the integrity of the endosomal membrane and targets the Golgi apparatus. The integrity of endosomal membranes was examined during MVM entry. A9 mouse fibroblasts were incubated with MVM and low molecular weight tetramethylrhodamine-labeled dextran (3 kDa). Mock-infected cells were used as controls. After 30 min, the cells were washed, trypsinized to remove bound but not internalized viruses, and processed for immunofluorescence to confirm internalization of MVM (Figure 6E; top panel). At progressive times after internalization, the release of dextrans from the endosomes into the cytosol was examined by confocal microscopy. Similar to B19V, dextran release was not detected at any time point in MVM-infected A9 cells, and their endocytic retention resembled that of mock-infected cells (Figure 6E), suggesting that the PLA2 of MVM is inactive in the endosomes.

Next, we investigated the involvement of the Golgi apparatus in MVM infection. To this end, the effect of BFA and GCA on MVM infection was tested. A9 cells were pretreated with either BFA or GCA for 30 min prior to infection. After 24 h, infectivity was measured by quantification of NS1 mRNA. Both drugs caused a significant reduction in infectivity when applied before infection and had a significantly lower effect when applied after 5 hpi (Figure 6F and G). These results suggest that the Golgi is also required for MVM infectious trafficking. However, unlike B19V, MVM infection was severely reduced in the presence of Retro-2, suggesting that MVM traffics to the Golgi following the canonical early endosomes-to-Golgi transport (Figure 6H). To explore the involvement of late endosome-to-Golgi transport, Rab9 was knocked down by RNAi in A9 cells. The reduction of Rab9 expression was confirmed by Western blot (Figure 6I). Rab9-knockdown A9 cells were infected with MVM, and NS1 mRNA was quantified 24 hpi by RT-qPCR. As shown in Figure 6J, Rab9 knockdown had no effect on MVM infection. Collectively, our data reveal common features in the cell entry mechanism of MVM and B19V, such as the lack of endosomal membrane damage and the requirement for a functional Golgi. However, the mechanism of retrograde transport to the Golgi differs between the two viruses.

## 4. Discussion

Globoside is a glycosphingolipid present on the plasma membrane of many cell types across different tissues [49]. Traditionally, globoside has been considered the primary cellular receptor for B19V [50]. However, the narrow tissue tropism and restricted uptake of B19V do not align with the ubiquitous expression of globoside. We identified the receptor-binding domain (RBD) of B19V in the N-terminal VP1u of B19V [24] and found that the VP1u cognate receptor, named VP1uR, is expressed exclusively in erythroid progenitor cells (EPCs), the only cells that permit efficient B19V internalization and productive infection [25,26]. A recent study identified the tyrosine protein kinase receptor AXL as an interacting partner of VP1u [51].

In previous studies, we confirmed that globoside does not serve as the primary cell surface receptor for B19V entry into EPCs; this role belongs to VP1uR. Instead, we found that globoside plays a critical role in a post-entry step for productive infection [27]. Further investigation revealed that B19V interacts with globoside exclusively under acidic conditions [28]. This pH-dependent interaction significantly impacts B19V infection, as the virus does not bind to the ubiquitously expressed globoside under neutral extracellular conditions. This strategy prevents globoside-mediated virus redirection to nonpermissive tissues and facilitates selective targeting of EPCs in the bone marrow. However, acidic niches in the body, such as the nasal mucosa, can become potential targets for B19V. In line with this, we demonstrated that the upper respiratory epithelium expresses globoside and allows virus attachment and transcytosis across the airways exclusively under the acidic pH of the nasal mucosa [23]. 

While globoside plays a central role in B19V transmission through the respiratory tract, its role in productive infection in bone marrow EPCs remains unclear. Strong evidence for the importance of globoside in the infectious trafficking of B19V comes from experiments with globoside KO cells. In cells expressing globoside, incoming capsids move from the endo-lysosomal compartment to a dispersed distribution with limited co-localization with endo-lysosomal markers. In globoside knockout cells, internalized particles remain associated with endo-lysosomal markers and the infection is blocked [28], suggesting that globoside plays a crucial role in the infectious trafficking of B19V.

In addition to the plasma membrane, globoside is also present in early endosomes and the trans-Golgi network (TGN), where it is synthesized from its precursor, globotriaosylceramide (Gb3) [52,53]. Under the acidic conditions characteristic of early endosomes and the TGN (pH 6.0–6.5), the affinity of the virus for VP1uR decreases, while its affinity for globoside increases. This pH-dependent shift in affinity likely facilitates the dissociation of incoming virus from the primary receptor VP1uR and promotes its binding to globoside in endosomes.

To better understand the endosomal entrapment of incoming viruses observed in globoside KO cells [28], the infection was performed in presence of PEI. We hypothesized that if globoside is required for endosomal escape, facilitating the release of viruses into the cytosol with PEI would overcome the endosomal entrapment of incoming virus. PEI functions as a proton sponge within endosomes. As the polycation sequesters protons, ATPase proton pumps persist in channeling protons into the endosomes. To maintain electrochemical balance, chloride ions are pumped into the endosomes, resulting in endosomal swelling and rupture [30,31]. Additionally, PEI can intercalate into the endosomal membranes, causing small local membrane damage [54]. B19V infection was blocked in globoside KO cells regardless of the MOI employed, confirming our previous study [28]. PEI-mediated endosomal leakage allowed the virus to circumvent the endosomal arrest imposed by the absence of globoside, confirming the role of intracellular globoside in facilitating endosomal escape. 

For most parvoviruses, the low endosomal pH environment, possibly in conjunction with endosomal proteases and/or autoproteolysis, induces conformational changes in the viral capsid, leading to the externalization of the VP1u-associated PLA2 motif [10,55]. The lipolytic activity of PLA2 is thought to cause membrane disruptions that allow the virus to access the cytosol [14]. The VP1u of B19V differs from other parvoviruses in that it is already exposed at the plasma membrane upon receptor binding [56,57]. However, B19V still requires the low endosomal pH [11], presumably to interact with globoside. Similar to globoside KO cells, B19V was arrested in endosomes in cells treated with bafilomycin A1 [11]. MS2 phages pseudotyped with B19V VP1u subunits, which can internalize UT7/Epo cells but cannot interact with globoside, were also arrested in the endosomes [28]. Taken together, these results suggest that when globoside interaction is impeded, incoming viruses are trapped in the endosomes and the infection is blocked.

Although the acidic environment in the endosomes is crucial to allow the interaction of the virus with globoside, it is not optimal for the PLA2 activity. B19V PLA2 is active in a narrow pH around 6.2, rapidly decreasing in acidic conditions. The concentration of Ca^2+^ is particularly important, and at concentrations under 20 μM, the activity is barely detectable. Calcium concentrations have been found to be much lower in the early endosome than in the extracellular milieu, with estimated calcium levels of ≈ 0.5 μM for Rab5a-positive endosomes and ≈2.5 μM for Rab7-positive endosomes, and increase during progression to the lysosome [58]. However, PLA2 is not active at the acidic pH of lysosomes (4.5–5.0). As a result, the viral PLA2 may not be active in the entire endolysosomal compartment. Consistent with this, B19V did not cause any detectable leakage of endocytosed dextrans during the entry process. Increasing the PLA2 potential of the virus by decorating phage particles with multiple active PLA2 constructs did not alter the results, strongly suggesting that endosomes are not the site of escape into the cytosol because the PLA2 is inactive or highly inefficient. This observation is in accordance with previous findings, where it was shown that canine parvovirus escapes from the endosomes without endosomal damage. Dextrans (10 kDa) and *α*-sarcin remained inside the vesicles during the endocytic trafficking of the virus [59,60]. This may well be the case for other parvoviruses, as we found that the PLA2 activity of MVM and PmPV decreases as the pH becomes increasingly acidic. 

Calcium concentration in the Golgi increases from the TGN to the cis-Golgi (130–250 µM, respectively) [38], and the pH decreases from cis-Golgi to the TGN (6.7–6.0, respectively) [39]. These conditions are better suited for the specific PLA2 enzymatic requirements. Accordingly, the Golgi apparatus represents an attractive target for the incoming virus. In line with this, it has been shown that AAV2 capsids accumulate in the Golgi apparatus shortly after internalization, and the integrity of the Golgi apparatus is important for infection, as treatment with BFA or GCA resulted in reduced transduction [15,16]. Similarly, B19V infection was blocked when the integrity of the Golgi was disrupted with BFA and GCA, but only when applied during entry. Interestingly, the virus was able to bypass non-functional Golgi by promoting endosomal escape with PEI, further emphasizing that the Golgi is required exclusively during entry. The higher calcium concentration in the cis-Golgi offers better enzymatic conditions for the viral PLA2 and may explain the stronger colocalization of incoming viruses in this compartment compared to the TGN. However, whether the virus escapes into the cytosol from the cis-Golgi or from any other compartment needs to be determined.

However, while disruption of the retromer complex by Retro-2 inhibits AAV infection [17], our study found it had no effect on B19V infection. Interestingly, we found that the infection with the parvovirus model MVM is also sensitive to Golgi disrupting agents, and similar to AAV, Retro-2 was also effective against the infection. Collectively, these results suggest that retrograde endosome-to-Golgi transport of B19V is mechanistically different and was not elucidated in this study. Toxins like Shiga and cholera and viruses such as SV40 and norovirus induce positive membrane curvature and budding by engaging in multivalent interactions with glycosphingolipids via pentameric structures [61,62,63,64]. Therefore, it is plausible that binding of globoside to the five-fold symmetry axis initiates a similar membrane invagination process, facilitating endosomal escape without causing membrane disruption. Consistent with this notion, in a previous study, we found that binding of B19V to globoside expressed on airway epithelial cells exposed to acidic conditions was required and sufficient to induce membrane invagination, leading to virus internalization [23]. 

In addition to its role as a receptor mediating viral transcytosis through the respiratory tract [23], this study identifies globoside as an intracellular receptor essential for B19V infectious entry and productive infection in erythroid progenitor cells. In conjunction with VP1uR, whose expression is restricted to erythroid progenitors, globoside additionally contributes to the marked erythroid tropism of B19V. Further functional and structural studies of the interaction between B19V and globoside are warranted, as they will provide mechanistic insights into viral transmission through the respiratory tract and intracellular trafficking in erythroid progenitor cells.

## Figures and Tables

**Figure 1 cells-13-01254-f001:**
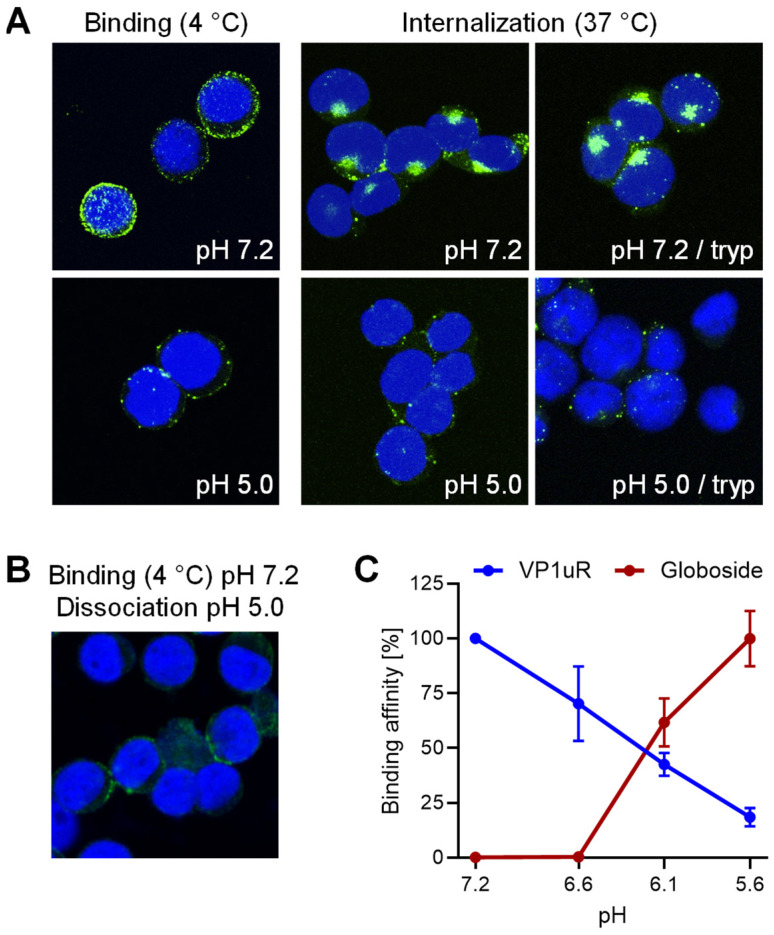
pH modulates the affinity of B19V for globoside and VP1uR. (**A**) Recombinant VP1u was allowed to interact with UT7/Epo cells at 4 °C for 1 h or at 37 °C for 30 min under neutral (pH 7.2) or acidic (pH 5.0) conditions. Cells were subsequently washed to remove unbound virus and fixed and stained for VP1u using an anti-Flag antibody. Where indicated, a trypsinization (tryp) step was performed to remove bound VP1u from the cell surface. (**B**) Recombinant VP1u was allowed to interact with UT7/Epo cells at 4 °C and neutral pH for 30 min followed by acidification (pH 5.0) and incubation for another 30 min. Cells were washed with ice-cold buffer at pH 5.0, fixed, and stained for VP1u using an anti-Flag antibody. (**C**) Binding affinity of B19V for globoside was determined by incubating purified B19V (3 × 10^9^) with RBCs (0.5%) in 100 µL buffer at pH ranging from 7.2 to 5.6 at RT for 1 h. Cells were washed with incubation buffer and DNA was extracted and quantified by qPCR. The binding affinity of VP1u for VP1uR was quantified by incubating PP7-VP1u (10^10^) with UT7/Epo cells in 100 µL buffer at pH ranging from 7.2 to 5.6 at 4 °C for 1 h. Cells were washed with incubation buffer and DNA was extracted and quantified by qPCR. Results are presented as the mean of two independent experiments ± standard deviation (SD).

**Figure 2 cells-13-01254-f002:**
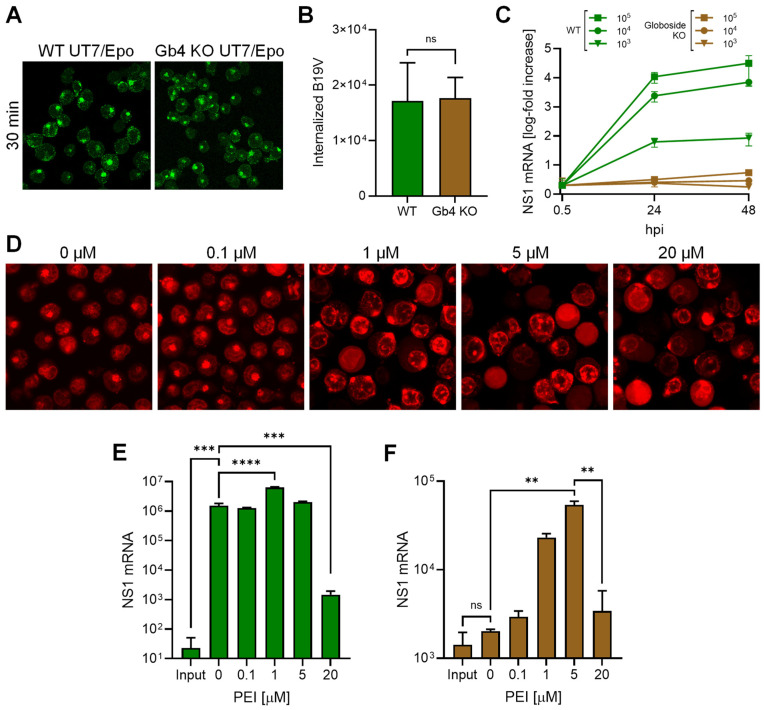
PEI-mediated endosomal leakage rescues B19V infection in globoside knockout cells. Internalization of B19V (10^5^ geq/cell) into WT and globoside (Gb4) KO UT7/Epo cells was analyzed by immunofluorescence using an antibody against intact capsids (**A**) and by qPCR (**B**). (**C**) WT and globoside KO UT7/Epo cells were infected with B19V (10^3^–10^5^ geq/cell). At the indicated time points, cells were harvested and washed, and viral NS1 mRNA was quantified by RT-qPCR. (**D**) UT7/Epo cells were incubated with tetramethylrhodamine-labeled dextran (3 kDa, 0.1 mg/mL) and increasing concentrations of PEI at 37 °C for 30 min. Cells were subsequently washed and live-cell imaging was performed using a confocal microscope. The effect of increasing concentrations of PEI on the infection of B19V (10^5^ geq/cell) in WT (**E**) and in globoside KO UT7/Epo (**F**) cells was examined. After 30 min, the virus and PEI were removed by a washing step, and infection was allowed to continue for up to 24 h, followed by RNA extraction and quantification of NS1 mRNA by RT-qPCR. Results are presented as the mean of three independent experiments ± standard deviation (SD). ns, not significant; **, *p* < 0.01; ***, *p* < 0.001; ****, *p* < 0.0001.

**Figure 3 cells-13-01254-f003:**
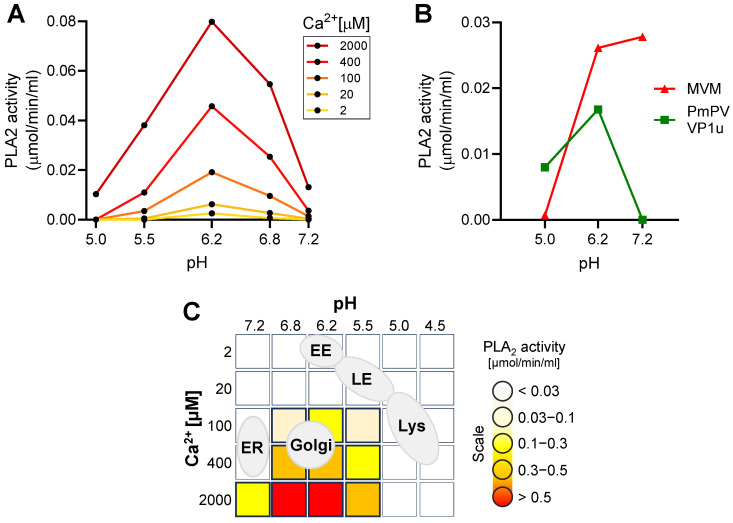
Identification of optimal enzymatic conditions for viral PLA2 activity. (**A**) The PLA2 activity of recombinant VP1u (1 µg) from B19V was assayed at pH 5.0–7.2 with increasing concentration of Ca^2+^. (**B**) The PLA2 activity of MVM (10^10^) and recombinant VP1u from PmPV (100 ng) was tested at pH 5.0–7.2 and 2 mM Ca^2+^. (**C**) Schematic depiction of different cellular organelles according to their pH and calcium concentrations superimposed on the optimal B19V PLA2 enzymatic conditions.

**Figure 4 cells-13-01254-f004:**
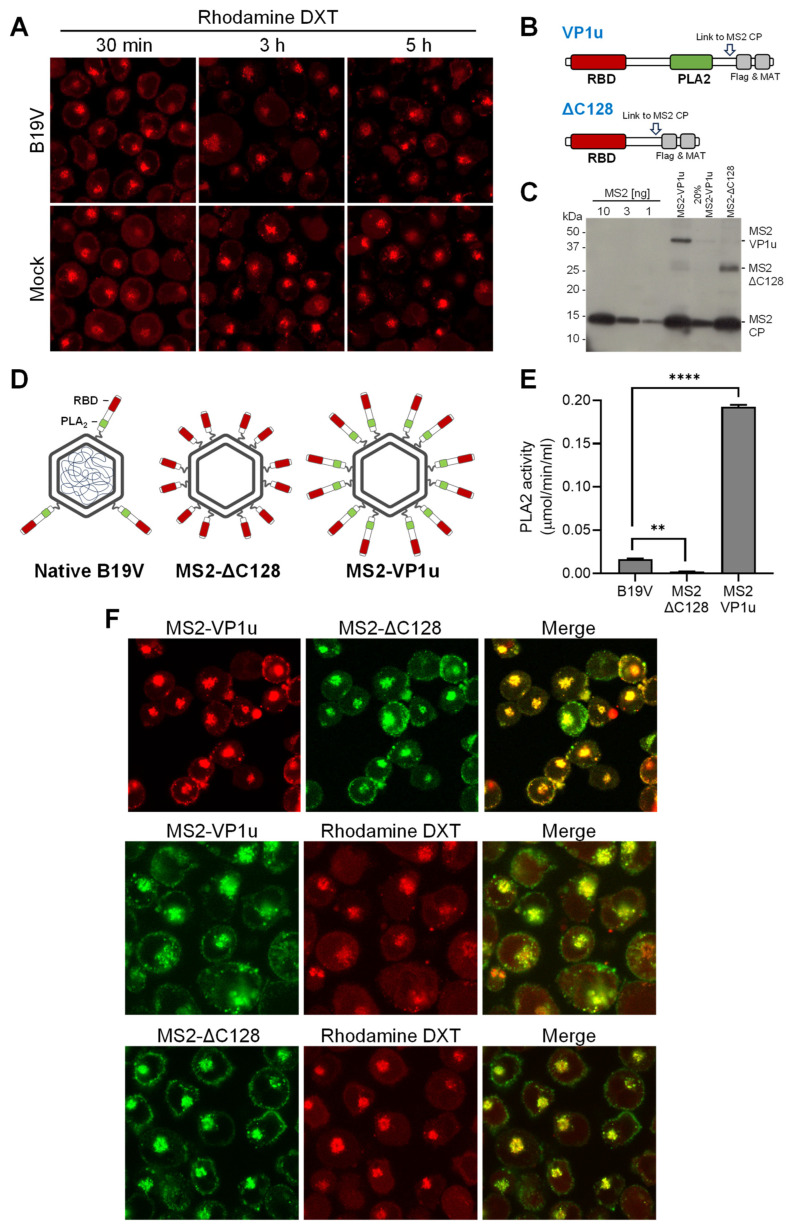
The PLA2 activity of B19V does not cause endosomal membrane damage. (**A**) UT7/Epo cells were incubated with B19V (10^5^ geq/cell) and tetramethylrhodamine-labeled dextran (3 kDa, 0.1 mg/mL) at 37 °C for 30 min. Cells were washed and incubated for up to a total of 5 h. Dextran (DXT) retention within the endosomes was analyzed by live-cell confocal microscopy. (**B**) Schematic depiction of full-length VP1u and ΔC128 VP1u used for coupling to MS2 capsids through a cysteine side chain (arrow). (**C**) MS2, MS2-VP1u, and MS2-ΔC128 were separated on an SDS-PAGE, transferred to a PVDF membrane, and detected with an anti-MS2 antibody. (**D**) Schematic representation of native B19V and MS2 constructs with a representative amount of VP1u. (**E**) The PLA2 activity of equal amounts of native B19V and MS2 particles was analyzed at pH 6.2 with 2 mM Ca^2+^. (**F**) Fluorescently labelled MS2-VP1u and MS2-ΔC128 (10^6^ geq/cell) were added to UT7/Epo cells with or without tetramethylrhodamine-labeled dextran (DXT, 3 kDa, 0.1 mg/mL) at 37 °C for 30 min, followed by a washing step to remove unbound particles and further incubated for 1 h. Live-cell imaging was performed using confocal microscopy to visualize dextrans and MS2 particles. Results are presented as the mean of two independent experiments ± standard deviation (SD). **, *p* < 0.01; ****, *p* < 0.0001.

**Figure 5 cells-13-01254-f005:**
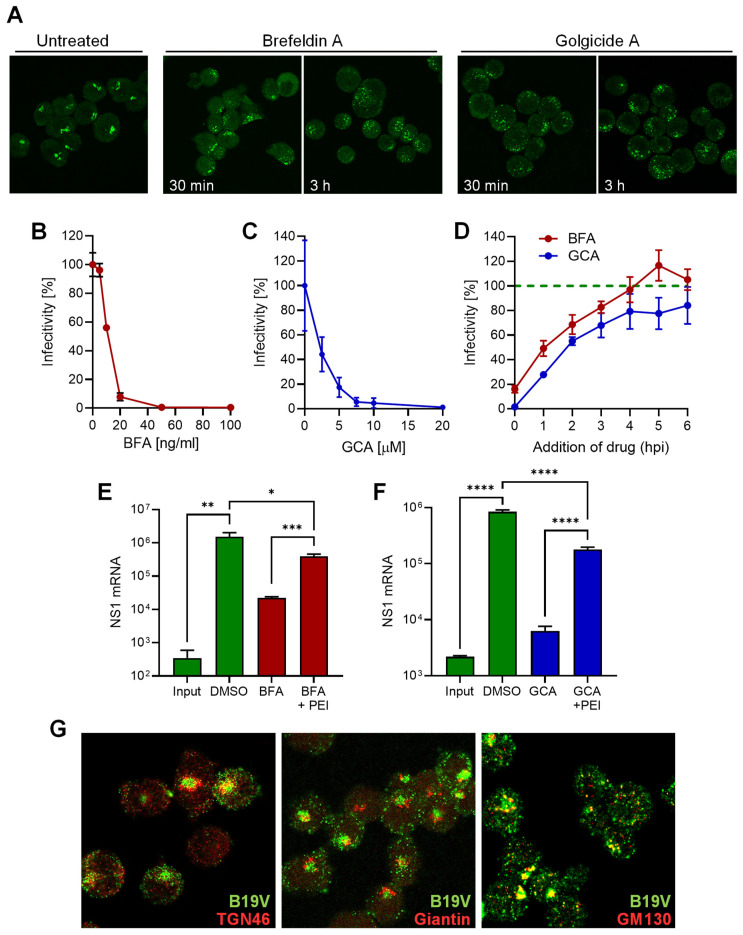
B19V entry requires a functional Golgi apparatus. (**A**) UT7/Epo cells were incubated with 50 ng/mL brefeldin A (BFA) or 20 µM golgicide A (GCA) for 30 min, 3 h, or left untreated. Cells were fixed and the Golgi apparatus was visualized with an anti-giantin antibody. Effect of increasing doses of BFA (**B**) or GCA (**C**) on B19V infection in UT7/Epo cells. Cells were pretreated with drugs for 15 min before addition of the virus. After 30 min at 37 °C, unbound virus was removed by a washing step and NS1 mRNA was quantified 24 hpi by RT-qPCR. (**D**) Effect of BFA (20 ng/mL) and GCA (10 µM) on B19V infection when added at progressive times post-infection compared to untreated cells (dotted green line). Effect of PEI on B19V infection in cell treated with BFA (**E**) or GCA (**F**). After drug treatment (BFA, 5 ng/mL; GCA, 10 µM) for 30 min, cells were infected with B19V (10^5^ geq/cell). When indicated, PEI (1 µM) was added at 37 °C for 30 min. Cells were then washed and incubated in medium containing the drugs. After 24 h, NS1 mRNA was quantified by RT-qPCR. (**G**) Co-localization of B19V with Golgi markers. UT7/Epo cells were infected with B19V (10^5^ geq/cell) at 37 °C for 30 min followed by a washing step to remove unbound virus. After 30 min, cells were fixed and stained with antibodies against intact capsids (green), and TGN46, giantin or GM130 (red). Results are presented as the mean of two (**B**–**D**) or three independent experiments for (**E**,**F**) ± standard deviation (SD). *, *p* < 0.05; **, *p* < 0.01; ***, *p* < 0.001; ****, *p* < 0.0001.

**Figure 6 cells-13-01254-f006:**
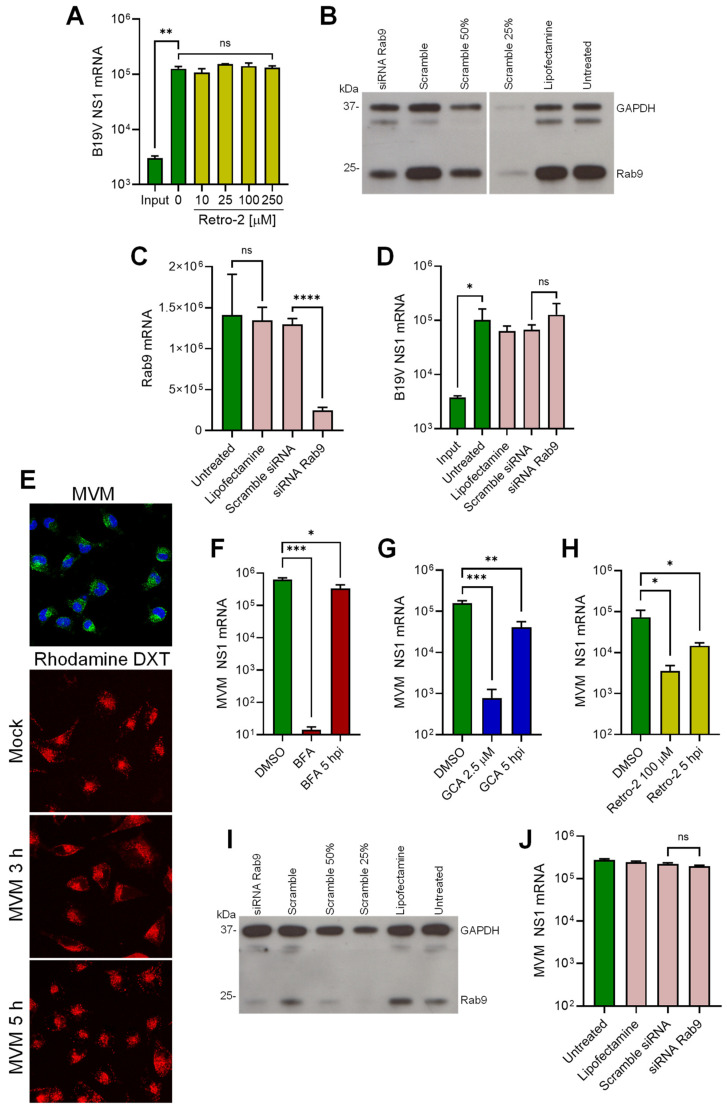
Endosome-to-Golgi retrograde transport of B19V and MVM. (**A**) UT7/Epo cells were treated with increasing concentrations of Retro-2 at 37 °C for 15 min before infection with B19V (10^5^ geq/cell). After 30 min, cells were washed to remove unbound virus and 24 hpi NS1 mRNA was quantified by RT-qPCR. Knockdown of Rab9 in UT7/Epo cells by RNAi. Cells were transfected with either Rab9 siRNA, a scrambled siRNA, mock-transfected (Lipofectamine), or left untreated. Two days after transfection, knockdown efficiency was assayed by (**B**) Western blot or (**C**) RT-qPCR. (**D**) B19V infection in Rab9 knockdown UT7/Epo cells. Rab9 knockdown cells were infected with B19V (10^5^ geq/cell), and NS1 mRNA was quantified 24 hpi by RT-qPCR. (**E**) Endosomal integrity in MVM-infected A9 cells. Cells were infected with MVM (2 × 10^3^ geq/cell) in the presence of absence of tetramethylrhodamine-labeled dextran (DXT, 3 kDa, 0.1 mg/mL) at 37 °C for 30 min. Cells were then either fixed and stained with mAb B7 against MVM capsids (upper panel) or incubated at 37 °C for up to 5 h and analyzed directly using fluorescence microscopy (lower panels). Effect of BFA (**F**), GCA (**G**) and Retro-2 (**H**) on MVM infection in A9 cells. Drugs were added 15 min prior to MVM infection or added 5 hpi. Unbound virus was removed after 30 min and NS1 mRNA was quantified 24 hpi by RT-qPCR. (**I**) Rab9 knockdown efficiency in A9 cells was quantified by Western blot as described in (**B**). (**J**) MVM infection in Rab9 knockdown A9 cells. Rab9 knockdown cells were infected with MVM (2 × 10^3^ geq/cell), and NS1 mRNA was quantified 24 hpi by RT-qPCR. Results are presented as the mean of two (**A**,**J**) or three (**C**,**D**,**F**–**H**) independent experiments ± standard deviation (SD). *, *p* < 0.05; **, *p* < 0.01; ***, *p* < 0.001; ****, *p* < 0.0001.

## Data Availability

Dataset available on request from the authors.

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
