# Peer review of "Globoside Is an Essential Intracellular Factor Required for Parvovirus B19 Endosomal Escape"

_cells, 2024, doi:10.3390/cells13151254_

Round 1
Reviewer 1 Report
Comments and Suggestions for Authors
This paper is a very nice continuation of the Authors‘ previous excellent work on parvovirus B19 entry mechanisms. It confirms their previous results concerning the switch of viral receptors depending on acidity and also adds considerable new knowledge that truly challenges the old theory of VP1u-associated phospholipase A2 (PLA2) to breach the endosomal membrane and release the B19V capsid into the cytosol. Instead, the Authors now give multiple controlled evidence of the viral receptor switch from a VP1u-binding receptor to globoside in the acidic endosome, which enables the virus to traverse to the much less acidic Golgi network, where PLA2 activity is much greater than in the acidic endolysosomal miljö. Contrary to earlier beliefs, they clearly showed that PLA2 does not work in the acidic endolysosomal conditions with low Ca2+ concentrations, but does work in the Golgi system — the maybe most interesting figure being Fig 3C. They further compared the B19V PLA2 with that of another (mouse) parvovirus, MVM, which partly showed the same phenomenon, but pointing to a slightly different mechanism for MVM trafficking, which does not involve globoside. The exact molecular mechanism on how globoside promotes this trafficking remains to be revealed.
The experiments are well designed, controlled and performed, and the findings support the conclusions. However, the Authors could clarify the following 6 minor issues in the text, which would help the readers‘ confusion:
1. Why was PmPV VP1u used instead of B19V, in the comparison to MVM.
2. What is the difference of VP1uR and AXL, is AXL the VP1uR? Now it was only stated that “AXL is an interacting partner of VP1u”?
3. This statement on line 411 is only partly true: “This strategy prevents virus redirection to nonpermissive tissues and facilitates selective targeting of EPCs in the bone marrow”. It would be good to mention that B19V does enter nonpermissive cells of many different tissues, and persist there for decades. The entry is, however, medited by antibodies and occurs via the FcR or complement factor C1q, but does not (presumably) result in productive replication.
4. These two sentences seem to contradict (lines 445-450): “Inside the endosomes, the low pH environment… induces conformational changes in the viral capsid, leading to the externalization of the VP1u-associated PLA2 motif” and “The VP1u of B19V differs from other parvoviruses in that it is already exposed at the plasma membrane upon receptor binding”.
5. Following the same thought: Clarify why also B19V capsids required heating to expose the VP1u (line 614)?
6. Very minor (section 4.4): “conjugated goat anti-human… what?” IgG? and “also later: “goat anti-mouse…?”.
Reviewer 2 Report
Comments and Suggestions for Authors
The authors have studied binding of VP1u as a recombinant peptide, and in constructs of PP7-VP1u and MS2-VP1u. The approach is justifiable, but it would be helpful for the authors to briefly discuss that VP1u as a peptide as well as in these constructs behaves as it would in native B19V virion.
A brief discussion of golgi apparatus trafficking would be helpful. Perhaps a better alternative would be a graphic figure illustrating the proposed B19V pathway would help the reader visualize the proposed pathway.
Line 402: the authors state that erythroid progenitor cells are "the only cells that permits internalization and infection." While erythroid progenitor cells may provide the cell type for productive infection, myeloid cells do allow viral production although at a lower level of replication efficiency. For example, see Srivastava A, Lu L. Replication of B19 parvovirus in highly enriched hematopoietic progenitor cells from normal human bone marrow. J Virol. 1988 Aug;62(8):3059-63. Further, the statement does not reflect clinical presentations in which other cell types are impacted, as in thrombocytopenia or leukopenia that may be seen in B19V infection.
Overall, well written and well reasoned.
Round 2
Reviewer 1 Report
Comments and Suggestions for Authors
The manuscript has been sufficiently improved.
Reviewer 2 Report
Comments and Suggestions for Authors
I appreciate the authors' responsiveness.